# Betaine Supplementation Improves 60 km Cycling Time Trial Performance and One-Carbon Metabolism in Cyclists During Recovery

**DOI:** 10.3390/nu17172765

**Published:** 2025-08-26

**Authors:** David C. Nieman, Camila A. Sakaguchi, James C. Williams, Jackie Lawson, Kevin C. Lambirth

**Affiliations:** 1Human Performance Laboratory, Appalachian State University, North Carolina Research Campus, Kannapolis, NC 28081, USA; olsonca1@appstate.edu (C.A.S.); williamsjc12@appstate.edu (J.C.W.); 2College of Computing and Informatics, University of North Carolina at Charlotte, Kannapolis, NC 28081, USA

**Keywords:** betaine, exercise, metabolomics, gut permeability

## Abstract

Background/Objectives: This study examined the effects of 2 weeks of betaine versus placebo supplementation (3 g/d) on 60 km cycling performance, gut permeability, and shifts in plasma metabolites. Methods: Participants included 21 male and female non-elite cyclists. A randomized, placebo-controlled, double-blind, crossover design was used with two 2-week supplementation periods and a 2-week washout period. Supplementation periods were followed by a 60 km cycling time trial. Six blood samples were collected before and after supplementation (overnight fasted state), and at 0 h, 1.5 h, 3 h, and 24 h post-exercise. Five-hour urine samples were collected pre-supplementation and post-60 km cycling after ingesting a sugar solution containing lactulose 5 g, ^13^C mannitol 100 mg, and ^12^C mannitol 1.9 g in 450 mL water. Other outcome measures included plasma intestinal fatty acid binding protein-1 (I-FABP), muscle damage biomarkers (serum creatine kinase, myoglobin), serum cortisol, complete blood cell counts, and shifts in plasma metabolites using untargeted metabolomics. Results: The time to complete the 60 km cycling bout differed significantly between the betaine and placebo trials (mean ± SE, 112.8 ± 2.3, 114.2 ± 2.6 min, respectively, (−1.41 ± 0.7 min) (effect size = 0.475, *p* = 0.042). No trial differences were found for I-FABP (interaction effect, *p* = 0.076), L:^13^CM (*p* = 0.559), the neutrophil/lymphocyte ratio (*p* = 0.171), serum cortisol (*p* = 0.982), serum myoglobin (*p* = 0.942), or serum creatine kinase (*p* = 0.694). Untargeted metabolomics showed that 214 metabolites exhibited significant trial treatment effects and 130 significant trial x time interaction effects. Betaine versus placebo supplementation was linked to significant increases in plasma betaine, dimethylglycine (DMG), sarcosine, methionine, S-adenosylhomocysteine (SAH), alpha-ketoglutaramate, and 5′methylthioadensone (MTA), and decreases in plasma carnitine and numerous acylcarnitines. Conclusions: Betaine supplementation modestly improved 60 km cycling performance but had no effect on gut permeability. The metabolomics data supported a strong influence of 2-week intake of betaine on the one-carbon metabolism pathway during the 24 h recovery period.

## 1. Introduction

Prolonged and intensive endurance exercise causes post-exercise inflammation, oxidative stress, immune suppression, and extensive perturbations in hundreds of circulating proteins and metabolites. Nutrition-based interventions are being investigated for their potential to improve long-distance running and cycling performance, mitigate physiological stress, and augment metabolic recovery [1,2,3,4,5].

Betaine (trimethylglycine) is a non-essential amino acid and is naturally present in beets, spinach, wheat germ and bran, and shellfish. Dietary intake ranges from 0.5 to 2.5 g/d depending on diet composition [6]. Betaine is also produced in vivo by means of choline metabolism. Betaine is one of several key methyl donors in the one-carbon metabolism pathway and participates in the methionine cycle by transferring a methyl group to homocysteine and contributing to the formation of S-adenosylmethionine (SAM) [7]. SAM is the universal methyl donor that supports histone and DNA methylation, two epigenetic mechanisms that regulate gene expression [8]. Betaine also functions as an organic osmolyte to maintain and regulate fluid balance, as well as intracellular fluid concentrations and cell volume [9,10]. Cell culture and animal studies suggest that betaine can reduce oxidative damage, improve enterocyte health, and counter negative changes in intestinal permeability [7,11]. Animal data indicate that betaine helps regulate intestinal epithelial barrier function by improving the intestinal barrier structure and the expression of tight junction proteins, including occludin and claudin-1 [12,13,14]. Plasma betaine concentrations are inversely associated with cardiovascular events [15], and higher levels have been linked to a pattern of favorable lifestyle habits and leanness [16].

Limited and conflicting data indicate a potential effect of betaine supplementation on human exercise performance and recovery [17,18]. The underlying mechanisms are poorly understood but may involve beneficial influences on fat oxidation, regulation of gene expression, growth hormone and insulin-like growth factor-1 (IGF-1) secretion, creatine and protein synthesis, regulation of inflammation and oxidative stress, cortisol production, regulation of organic osmolytes and cell water retention, and sensations of fatigue [17,18]. These betaine-related physiological effects suggest a potential influence on performance and metabolic recovery from both resistance exercise and long-distance cycling and running. However, most betaine supplementation studies have focused on strength-related outcomes or short bursts of intense exercise [18,19,20,21,22]. A recent review of 17 studies concluded that betaine supplementation for at least 7 days may improve various measures of muscular strength [18]. In this systematic review and meta-analysis, a significant effect size of 0.47 was reported for maximal strength in the lower body but not for upper body strength, cycling sprint power, or muscular endurance [18]. Few studies have investigated betaine’s influence on long-distance cycling and running performance and metabolic recovery. One study reported no effects of 2-week intake of betaine supplements (2.5 g/d) on intensive running time to exhaustion (~33 min) and post-exercise oxidative stress, with a modest effect in lowering lymphocyte apoptosis [2]. Another study showed no performance effect of acutely ingesting 5 g betaine in one liter of rehydration beverages prior to 75 min of treadmill running followed by a performance sprint to exhaustion (3–4 min at 84% VO2max) [23].

The impact of betaine supplementation on long-distance cycling performance, gut permeability, and metabolic recovery has not been investigated. We hypothesized that ingestion of 3 g/d betaine versus placebo for two weeks in trained cyclists prior to a 60 km cycling time trial would improve performance, moderate exercise-induced changes in gut permeability, and improve metabolic recovery using a randomized crossover trial design. Outcome measurements also included the Profile of Mood States (POMS) to determine betaine’s potential influence on fatigue, vigor, and other mood states. A moderate dose of betaine (3 g/d) for a 2-week period was chosen for this investigation and represents the upper level of dietary intake achievable by individuals with diets high in whole grains, spinach, beets, and seafood [6]. The ergogenic and clinical effects of betaine have been investigated with doses ranging from 0.5 to 20 g/d, with 1.5–6 g/d found to be sufficient to increase plasma betaine and methionine concentrations and lower homocysteine within 2 weeks [6].

## 2. Materials and Methods

Study Participants

Healthy male and female cyclists were invited to take part in this study if they met the inclusion criteria. These included the following: 18 to 65 years of age, capable of cycling 60 km, and a willingness to restrict intake of betaine-rich foods (beets, spinach, wheat bran, and wheat germ) and avoid betaine supplements, other supplements, and nonsteroidal anti-inflammatory drugs or NSAIDs. To be included in the study, study participants had to report that they did not have a gastrointestinal disease (irritable bowel syndrome, chronic nausea, vomiting and diarrhea, Crohn’s disease, Celiac disease, diverticulosis) or chronic disease.

A total of 41 participants were assessed for eligibility, and 23 were entered into the study, with 21 completing all aspects of the protocol (Figure 1). A post hoc analysis using repeated measures for between trials and timepoints showed a statistical power of 0.93 for plasma betaine and dimethylglycine (two metabolites of interest in this study) at an alpha level of 0.05 with 21 subjects (WebPower, R package v. 0.9.4). Participants voluntarily signed the informed consent, and procedures were approved by the university’s Institutional Review Board. Trial Registration: ClinicalTrials.gov, U.S. National Institutes of Health, identifier: NCT06392360.

Study Design

This study employed a randomized, placebo-controlled, double-blind, crossover design with two 2-week supplementation periods (betaine and placebo capsules) and a 2-week washout period. The 2-week supplementation period was utilized because this duration was similar to what was used in 13 of 17 studies in a recent review. The supplement capsules were supplied in coded bottles by the sponsor, with the double-blind code held until after all study samples had been analyzed.

Study procedures were conducted at the Appalachian State University Human Performance Laboratory (HPL), North Carolina Research Campus, Kannapolis, NC. Some of the methods related to the research design, exercise protocol, and gut permeability were similar or adapted from a recent study published by our research group [24]. The reader is referred to reference [24] for additional details.

Subjects came to the lab for orientation/baseline testing, pre- and post-2-weeks supplementation blood sample collections (overnight fasted state), and two 60 km cycling sessions. Study participants signed the consent form and completed the delayed onset of muscle soreness (DOMS) [25] and Profile of Mood States (POMS) [26] questionnaires.

The BodPod system (Cosmed, Rome, Italy) and the seca medical Body Composition Analyzer (mBCA) (seca, Chino, CA, USA) were used to assess body composition, total body water, and extracellular water. Urine was collected for 5 h after ingestion of a nonabsorbable sugar solution (SS) containing 5 g lactulose (Sigma Aldrich, St. Louis, MO, USA), 100 mg ^13^C mannitol (Cambridge Isotope Laboratories, Tewksbury, MA, USA), and 1.9 g ^12^C mannitol (Sigma Aldrich) in 450 mL water [24]. Gut permeability was calculated from the post-exercise urine lactulose/^13^C mannitol ratio (L:^13^CM) [15].

Maximal aerobic capacity (VO_2max_) was assessed using a graded cycling test [24]. A 2-week supply of the coded supplement capsules (either betaine or placebo) was given to the participants, with instructions to ingest 6 capsules per day (3 g/d, 500 mg/capsule), with 3 capsules ingested daily with the first meal in the morning and 3 more with the last meal of the day. The betaine and placebo capsules were prepared by the study sponsor (AGRANA Beteiligungs-AG, Vienna, Austria). The white DuraBeet^®^ crystalline betaine was produced from beet molasses at the betaine crystallization plant in Austria according to European standards. The placebo capsules each contained 355 mg microcrystalline cellulose, 3 mg magnesium stearate, and 2 mg precipitated silica in identical-looking capsules. Participants were instructed to swallow the capsules whole with a beverage and not bite down on them. The coded supplement bottles were brought back to the lab after 2 weeks to help verify compliance with the supplementation regimen.

Participants tapered exercise training and ingested a moderate-carbohydrate diet before the 60 km exercise trials. Nutrient intake was calculated using the Food Processor dietary analysis software system (Version 11.11, ESHA Research, Salem, OR, USA).

After the 2-week supplementation period, study participants engaged in a 60 km cycling time trial. Blood samples, DOMS and POMS ratings, and seca bioelectrical impedance (BIA) measurements were obtained before and after the cycling sessions. Participants ingested three betaine or placebo supplement capsules with 500 mL of water before cycling. Participants cycled for 60 km at race pace intensity on their own bicycles fitted to Saris H3 direct drive smart trainers (Saris, Madison, WI, USA) with metabolic monitoring using the Cosmed CPET metabolic cart (Cosmed, Rome, Italy) [24]. Blood samples were collected immediately after and 1.5 h, 3.0 h, and 24 h post-exercise. Urine was collected for 5 h after ingesting the 450 mL SS [24]. Participants ingested three betaine or placebo supplement capsules with the last meal of the day.

Participants completed a 2-week washout period without the supplements, crossed over to the opposite treatment arm, and then repeated all procedures.

Sample Analysis

Plasma aliquots were prepared from EDTA blood collection tubes and stored in a −80 °C freezer for metabolomics and intestinal fatty acid binding protein (I-FABP) analysis. Serum creatine kinase, myoglobin, cortisol, and complete blood counts (CBCs) (EDTA tubes) were analyzed using Labcorp services (Labcorp, Burlington, NC, USA).

Urine Sugar and Plasma I-FABP Analysis:

The urine samples were analyzed using a high-performance liquid chromatography (HPLC) method for ^12^C- and ^13^C-mannitol and lactulose at the Mayo Clinic’s Immunochemical Core Lab (Mayo Clinic, Rochester, MN, USA) [27,28]. See reference [24] for a complete description.

Plasma Untargeted Metabolomics Analysis:

Plasma samples were analyzed using untargeted metabolomics procedures at Metabolon (Metabolon, Morrisville, NC, USA) [29]. Samples were prepared using the automated MicroLab STAR^®^ system from Hamilton Company (Hamilton Company, Reno, NV, USA). Several recovery standards were added prior to the first step in the extraction process for quality control (QC) purposes. Proteins were precipitated with methanol under vigorous shaking (Glen Mills GenoGrinder 2000, Glen Mills, Clinfton, NJ, USA), followed by centrifugation. The resulting extract was divided into multiple fractions: two for analysis by two separate reverse phase (RP)/UPLC-MS/MS methods with positive ion mode electrospray ionization (ESI), one for analysis by RP/UPLC-MS/MS with negative ion mode ESI, and one for analysis by HILIC/UPLC-MS/MS with negative ion mode ESI, while the remaining fractions were reserved for backup. Samples were placed briefly on a TurboVap^®^ (Zymark, Happy Valley, OR, USA) to remove the organic solvent. Several types of QC controls were analyzed in concert with the experimental samples: a pooled matrix sample, extracted water samples, and a cocktail of QC standards. Instrument variability was determined by calculating the median relative standard deviation (RSD) for the standards that were added to each sample prior to injection into the mass spectrometers. Overall process variability was determined by calculating the median RSD for all endogenous metabolites present in 100% of the pooled matrix samples. Experimental samples were randomized across the platform run with QC samples spaced evenly among the injections. All methods utilized a Waters ACQUITY ultra-performance liquid chromatography (UPLC) and a Thermo Scientific Q-Exactive high resolution/accurate mass spectrometer interfaced with a heated electrospray ionization (HESI-II) source and Orbitrap mass analyzer operated at 35,000 mass resolution. Raw data was extracted, peak-identified, and QC-processed using a combination of Metabolon-developed software services. Metabolon maintains a library based on authenticated standards that contains the retention time/index (RI), mass to charge ratio (*m*/*z*), and fragmentation data on all molecules present in the library. Peaks were quantified using area-under-the-curve. Data was normalized by the medians equal to one method and normalizing each data point proportionately.

Statistical Procedures

Data are expressed as mean ± SE. Except where described, data sets were analyzed using the generalized linear model (GLM) and the repeated measures ANOVA module in SPSS (IBM SPSS Statistics, Version 28.0, IBM Corp, Armonk, NY, USA). The statistical model utilized the within-subjects approach: 2 (supplement) (betaine or placebo) × 6 (timepoints) (pre-and post-2 weeks supplementation, and 0 h, 1.5 h, 3 h, and 24 h post-exercise) repeated measures ANOVA and provided time (i.e., the collective effect of the cycling exercise bout), supplement (i.e., the collective supplement effect), and interaction effects (i.e., whether the data pattern over time differed between supplement trials). If the interaction effect was significant (*p* ≤ 0.05), then post hoc analyses were conducted using paired *t*-tests comparing timepoint contrasts between supplement trials. An alpha level of *p* < 0.05 was used for statistical significance testing for paired *t*-tests (non-repeated supplement trial comparisons). A *p*-value of ≤0.0125 was used after Bonferroni correction for 4 multiple tests of changes from pre-study trial comparisons within repeated measures ANOVA analyses.

The metabolomics data was analyzed statistically using the ArrayStudio/Jupyter Notebook on log transformed data. Statistical analyses included principal component analysis (PCA) and two treatment two period crossover ANOVA. *p*-values and q-values were used for statistical significance testing. Partial Least Squares Discriminant Analysis (PLS-DA) was used [1] to identify metabolites that were discriminatory between the betaine and placebo trials, and [2] for differentiation between all six timepoints (ropls R package v.1.38.0). Metabolites with Variable Importance in the Projection scores (VIPs) were considered to be significant if ≥2.5. VIPs were cross-validated using features produced from Lasso regression with an expected binary outcome (alpha = 0.7) (caret R package v.7.0-1). Model performance was evaluated using cross-validation inherent within the function for 7 iterations. The resultant cumulative Q2 and proportion of the variance in response to the treatment or timepoint (R2Y) values were used to assess model predictability and quality. The effect of sex on metabolite changes was determined using Student’s t-test or a non-parametric equivalent between the sexes with all timepoint measurements. A two-way repeated-measures ANOVA was also utilized to check for interactions between sex and treatment. The steps for t-test analysis included assessing statistical power using Cohen’s D effect size, checking the assumptions of normality (Shapiro–Wilk) and equal variances (F-test), and then applying the appropriate test (Student’s *t*-test or Wilcox). ANOVA procedures included 0–3 h post-exercise timepoints and all timepoints. Statistical power was assessed for two-way ANOVA using Cohen’s F effect size for each group and checked using assumptions of normality, including Mauchly’s test. Greenhouse–Geisser *p*-value corrections were made if the assumption of sphericity was not met. FDR corrections for *p*-values for the main tests were made using Benjamini–Hochberg.

## 3. Results

A total of 21 study participants (*n* = 15 males, *n* = 8 females) completed all study procedures (Table 1). Age, BMI, percent body fat, and maximal heart rates were comparable between sex groups, with higher body mass, cardiorespiratory capacity (VO_2max_), maximal cycling power, and ventilation capacity for the male versus female cyclists (*p* < 0.05). The pattern of change over time did not differ between the male and female cyclists for urine lactulose/^13^C mannitol ratio (L:^13^CM) (supplement × time × sex interaction effect, *p*-value =0.634, plasma I-FABP, interaction *p*-value = 0.461), or plasma betaine (interaction *p*-value = 0.852). No sex differences were found for 60 km cycling performance in terms of average percent maximal heart rate, power output in watts, and oxygen consumption rates (all *p* > 0.13). In both types of ANOVA analyses, there were no statistically significant differences found between supplement trials examining all metabolites, and no interactions between supplement and gender were found. Thus, outcome measures are presented for all study participants combined.

Performance data for the 60 km cycling time trials after 2 weeks of supplementation with betaine and placebo are summarized in Table 2. The time to complete the 60 km cycling time trials was faster with betaine compared to placebo (−1.41 ± 0.7 min) (effect size d = −0.475, 95% CI [−0.955, 0.005], *p* = 0.042). No supplement trial differences were found for average power output, speed, oxygen consumption rates, respiratory exchange ratio (RER), rating of perceived exertion (RPE), and heart rates.

Three-day food records were collected at the end of each 2-week supplementation period. No differences in macro- and micro-nutrient intake were found between trials. Intake during the betaine and placebo trials did not differ for folate (355 ± 53.9, 358 ± 56.4 µg/d, *p* = 0.956) and choline (344 ± 33.3, 315 ± 28.2 mg/d, *p* = 0.342), respectively. Nutrient data from the two 3-day food records were averaged for the 21 cyclists: energy intake averaged 2117 ± 122 kcal/d (8.86 ± 0.51 MJ/d), and carbohydrate, protein, fat, and alcohol represented 42.0 ± 1.7, 20.6 ± 1.3, 36.8 ± 1.4, and 1.4 ± 0.5%, respectively, of total energy. Symptom logs revealed no significant trial differences.

No trial differences were found for plasma I-FABP (time effect, *p* = 0.013, interaction effect, *p* = 0.076) (Figure 2), L:^13^CM (time *p* = 0.237, interaction *p* = 0.559) (Figure 3), or L:^12^CM (time *p* = 0.071, interaction *p* = 0.695). Plasma volume shifts did not differ between the placebo and betaine trials immediately post-exercise (−7.7 ± 1.2% and −7.3 ± 0.8%, respectively, *p* = 0.800) and 1.5 h post-exercise (1.6 ± 0.9% and 2.3 ± 1.2%, respectively, *p* = 0.637).

The seca BIA data showed that the pattern of change over time differed for total body intracellular water between the betaine and placebo trials (interaction effect, *p* = 0.010), with changes from pre-supplementation levels decreasing in the betaine group compared to placebo (Table 3). No trial differences were found for total body extracellular water (interaction effect, *p* = 0.640).

The plasma metabolomics dataset after sample analysis included 1483 biochemicals with 1214 compounds of known identity and 269 compounds of unknown structural identity. Of the 1483 detected biochemicals, 1278 exhibited significant time-dependent effects. Thus, the 60 km cycling time trial caused an extensive perturbation in circulating metabolites. Statistical analysis showed that 214 metabolites exhibited significant trial treatment effects, and 130 metabolites had significant trial x time interaction effects. Appendix A include the raw and batch normalized values for each sample and the statistical analysis results in the form of a heatmap.

The Partial Least Squares Discriminant Analysis (PLS-DA) for the betaine and placebo trials across all six timepoints (R2Y = 0.495, Q2 = 0.451) is shown in Figure 4a (T1 = presupplementation, T2 = post-2-weeks supplementation, T3 = immediately post-exercise, T4 = 1.5 h post-exercise, T5 = 3 h post-exercise, T6 = 24 h post-exercise). The 60 km cycling time trial had a significant effect on metabolite shifts, especially for the first three hours of recovery. Figure 4b shows the PLS-DA contrast for the betaine and placebo trials (all six timepoints). A distinct difference was shown between the betaine and placebo trials (R2Y = 0.769, Q2 = 0.574). Lasso regression at a level of 0.7 and Variable Importance in Projection (VIP) scores of 2.5 and higher identified nine metabolites that were most influential in distinguishing the betaine and placebo trials, including plasma betaine, dimethylglycine (DMG), sarcosine, carnitine, deoxycarnitine, 5′methylthioadensone (MTA), acetylcarnitine (C2), alpha-ketoglutaramate, and N-acetylserine.

Betaine versus placebo supplementation was linked to significant increases in plasma betaine (treatment and timepoint interaction effect, q < 0.001) and the betaine metabolites dimethylglycine (DMG) (q < 0.001), and sarcosine (q < 0.001) (Figure 5a–c). Betaine supplementation was also associated with significant increases in methionine (q = 0.036) and related metabolites S-adenosylhomocysteine (SAH) (q = 0.025), alpha-ketoglutaramate (q < 0.001), and 5′methylthioadensone (MTA) (q < 0.001) (Figure 6a–d). Betaine supplementation was also related to a decrease in plasma carnitine (q < 0.001) (Figure 7) and numerous acylcarnitines (Appendix A).

Significant post-exercise increases (all *p* < 0.001) were measured for plasma inflammation-related metabolites, including 13-HODE + 9-HODE, 9,10- and 12,13-DiHOMEs, and 12-HETE. The pattern of increase over time for each of these metabolites, however, did not differ between the betaine and placebo trials (all *p* > 0.05) (Appendix A).

## 4. Discussion

This study showed that the intake of a modest 3 g/d dose of betaine for a relatively short 2-week supplementation period improved 60 km cycling time trial performance. Betaine supplementation was also linked to significant increases in pre- and post-exercise plasma levels of betaine and related metabolites in the one-carbon metabolism pathway, with reduced levels of carnitine and acylcarnitines. No effect of betaine supplementation was found for gut permeability, I-FABP, muscle soreness or muscle damage biomarkers, serum cortisol, exercise-induced mood disturbance, or inflammation biomarkers. Contrary to expectations, betaine supplementation was linked to a small but significant decrease in intracellular water.

This is the first study to show that betaine supplementation improves prolonged and intensive cycling performance. Most previous studies in this area focused on strength and power performance outcomes, and systematic reviews from 2017 and 2024 indicated moderate to high study heterogeneity and an inconsistent or modest betaine supplementation effect on lower body strength [18,20]. Proposed mechanisms include betaine-related effects on increased plasma levels of anabolic hormones, decreased cortisol responses, increased creatine and protein synthesis, improved cell water retention, and decreased feelings of fatigue [18,20,30]. However, data to support these suppositions are inconsistent or lacking in randomized human trials. Our data showed a modest but significant effect of betaine supplementation on 60 km cycling time trial performance using a randomized, crossover design and double-blinded, placebo-controlled supplementation methods. The effect size of −0.475 falls into the medium category, meaning that the observed cycling trial performance difference is evident but not necessarily very strong. No trial differences were found for exercise-induced muscle soreness or damage, inflammation, changes in serum cortisol, total mood disturbance (an index that includes fatigue), or shifts in plasma volume. Contrary to what has been proposed, BIA data supported a small but significant decrease in post-exercise intracellular water with betaine supplementation.

Enhanced cycling performance with betaine supplementation may in part be related to increases in the methyl donor pool that lead to favorable epigenetic influences on gene expression within muscle tissue [18]. The elevated pre- and post-exercise plasma levels of betaine, methionine, dimethylglycine, and sarcosine with betaine supplementation confirm its efficacy in influencing the one-carbon metabolism pathway. One-carbon metabolism is significantly involved in epigenetic modifications by providing methyl groups for DNA methylation, impacting gene expression [8]. Methionine plays a pivotal role in multiple biological functions and serves as a protein building block, as well as functioning as a substrate/precursor for several important downstream metabolites, including the universal methyl donor S-adenosylmethionine (SAM) and the antioxidant glutathione. Although SAM was not detected in this study using untargeted metabolomics, its post-methyl-donating product, S-adenosylhomocysteine (SAH), was significantly and robustly elevated following betaine supplementation. This suggests that betaine supplementation promotes one-carbon metabolism by efficiently re-methylating homocysteine to methionine, which can then be used to generate additional methyl-donating SAM [31]. Animal studies support that elevations in SAM via betaine supplementation promote muscle protein synthesis and improved grip strength through a regulatory effect on the mammalian target of rapamycin complex1 protein kinase (mTORC1) [32].

Prolonged and intensive endurance exercise evokes extensive physiological adaptive responses through transcriptional reactions that are associated with epigenetic DNA methylation and histone modifications [33,34]. There is increasing evidence that physical activity and nutrition work together as major environmental factors to shape human genotypes and phenotypes [35]. Whether or not an increase in the DNA methyl donor pool through betaine supplementation amplifies and facilitates the epigenetic regulation of gene transcription for exercise-induced adaptations has not yet been investigated in humans. A study with young and old mice showed that betaine supplementation transcriptionally improved age-related mitochondrial respiration in skeletal muscle and improved running distance to exhaustion [36]. A cell culture study with murine myoblasts showed that betaine stimulated muscle fiber differentiation by promoting the expression profile of genes encoding proteins involved in the IGF-1 pathway [37].

Animal studies support decreases in fat mass and reduced intramyocellular lipid accumulation with betaine supplementation [38,39]. A systematic review of human studies showed a modest fat-lowering effect with higher intakes of betaine [40]. The downward shift observed in circulating carnitine concentrations with betaine supplementation may represent a translocation of circulating carnitine into organ-bound liver and muscle carnitine. Our results are consistent with a recent metabolomics-based investigation of 32 dogs that showed a decrease in many plasma carnitine-containing metabolites with betaine supplementation [41]. Betaine acts as a methyl donor in the biosynthesis of carnitine, suggesting a positive impact on fat metabolism by enhancing fatty acid transport into the mitochondria through increased carnitine availability. However, our RER data did not support this potential glycogen-sparing effect within an exercise context. Additionally, plasma levels of major lipid oxidation intermediates formed during exercise (see Appendix A) were similar across the betaine and placebo arms, suggesting that lipid oxidation rates during exercise were not significantly impacted by betaine supplementation. The performance and physiological effects of betaine’s influence in lowering plasma levels of carnitines and acylcarnitines remain to be determined.

Betaine is an osmolyte that helps maintain fluid balance and intracellular fluid volumes [9,10]. However, the data from our study showed a small but significant decrease in intracellular water with betaine supplementation. The 8-point BIA method used in our study is considered an accurate and reliable method for estimating total body, extracellular, and intracellular water [42]. Other exercise-related studies using similar methods reported no effect of betaine supplementation on total body water [43]. Body water responses to betaine supplementation may depend on the dosing regimen and the degree of physiological stress imposed during the exercise intervention. Further research is needed using sophisticated fluid balance and diet control methods to determine the potential effects of betaine supplementation on intracellular water and performance in human athletes.

Betaine supplementation had no significant effect on the modest exercise-induced changes in I-FABP and gut permeability. Animal data indicates that betaine may influence gut permeability by regulating intestinal epithelial barrier function and structure [11,12,13,14,44]. The underlying mechanisms may include betaine’s protective effect on epithelial cell health and the upregulation of gene expression for tight junction proteins like occludin and zonula occludens-1, especially under conditions of heat or oxidative stress [44]. A limitation in this study was that changes in gut permeability were modest following the 60 km cycling bout. One review concluded that vigorous endurance exercise lasting at least 60 min at 70% VO2max was a sufficient stimulus to induce increased intestinal permeability, but that this varies widely between participants [45]. Additional studies should focus on similar exercise bouts combined with environmental heat stress [11]. Other limitations in this study included the lack of epigenetic and cytokine measurements to determine the potential influence of betaine supplementation on gene expression within muscle tissue and post-exercise inflammation. The betaine dosing regimen used in this study was similar to that used in most other related studies [18]. The potential influence of a larger betaine dose for a greater time period remains to be determined. We cannot rule out that the performance effect and metabolite shifts reported in this study were due in part to the acute intake of betaine just prior to the cycling time trial. This study utilized an acute 60 km cycling trial as a performance measurement, and the favorable results may not apply to other indices, including exercise time to exhaustion or chronic performance. The finding that betaine supplementation was linked to a small decrease in intracellular water was unexpected. This result may vary depending on the type of method used to assess fluid balance and intracellular fluid volumes.

## 5. Conclusions

Betaine supplementation did not influence several post-exercise secondary outcomes in these trained cyclists, including gut permeability, muscle damage, inflammation, serum cortisol levels, or mood state. Betaine compared to placebo supplementation was linked in this study to a modest improvement in 60 km cycling performance time and significant increases in circulating metabolites that are part of the methyl donor pool involved in various biological reactions within cells. These are novel data, and while promising, research on betaine’s potential role in exercise epigenetics is still in its early stages. The data from this investigation can form the basis for additional studies to determine if betaine supplementation contributes to improved exercise performance, recovery, and adaptation by modulating epigenetics through methylation.

## Figures and Tables

**Figure 1 nutrients-17-02765-f001:**
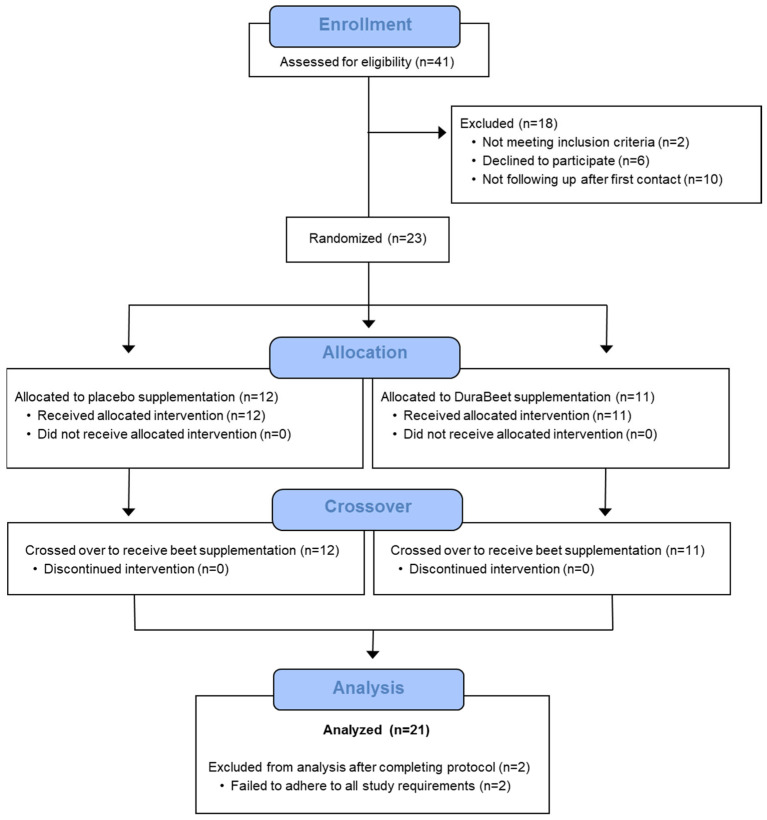
Study participant flow diagram.

**Figure 2 nutrients-17-02765-f002:**
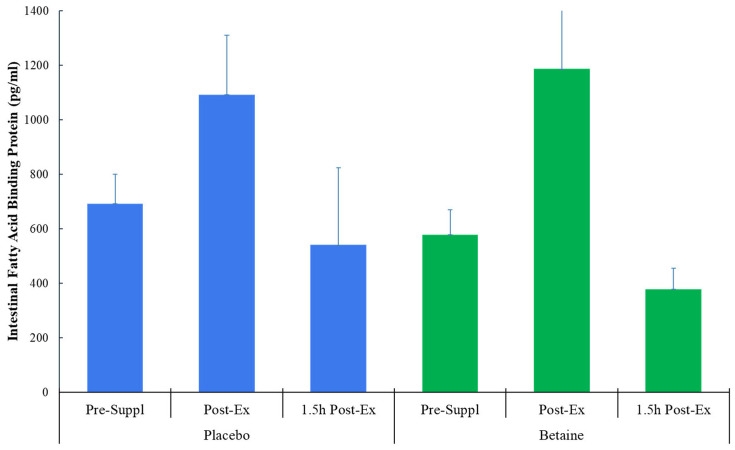
Betaine and placebo trial changes in intestinal fatty acid binding protein (I-FABP) across three timepoints (pre-supplementation, immediately, and 1.5 h post-exercise (60 km cycling time trial)).

**Figure 3 nutrients-17-02765-f003:**
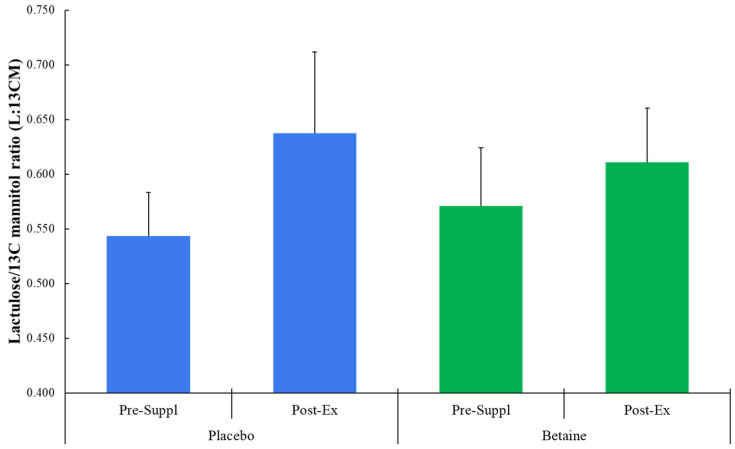
Betaine and placebo trial changes in lactulose to 13C mannitol ratios from 5 h urine samples collected pre-supplementation and post-exercise (60 km cycling time trial).

**Figure 4 nutrients-17-02765-f004:**
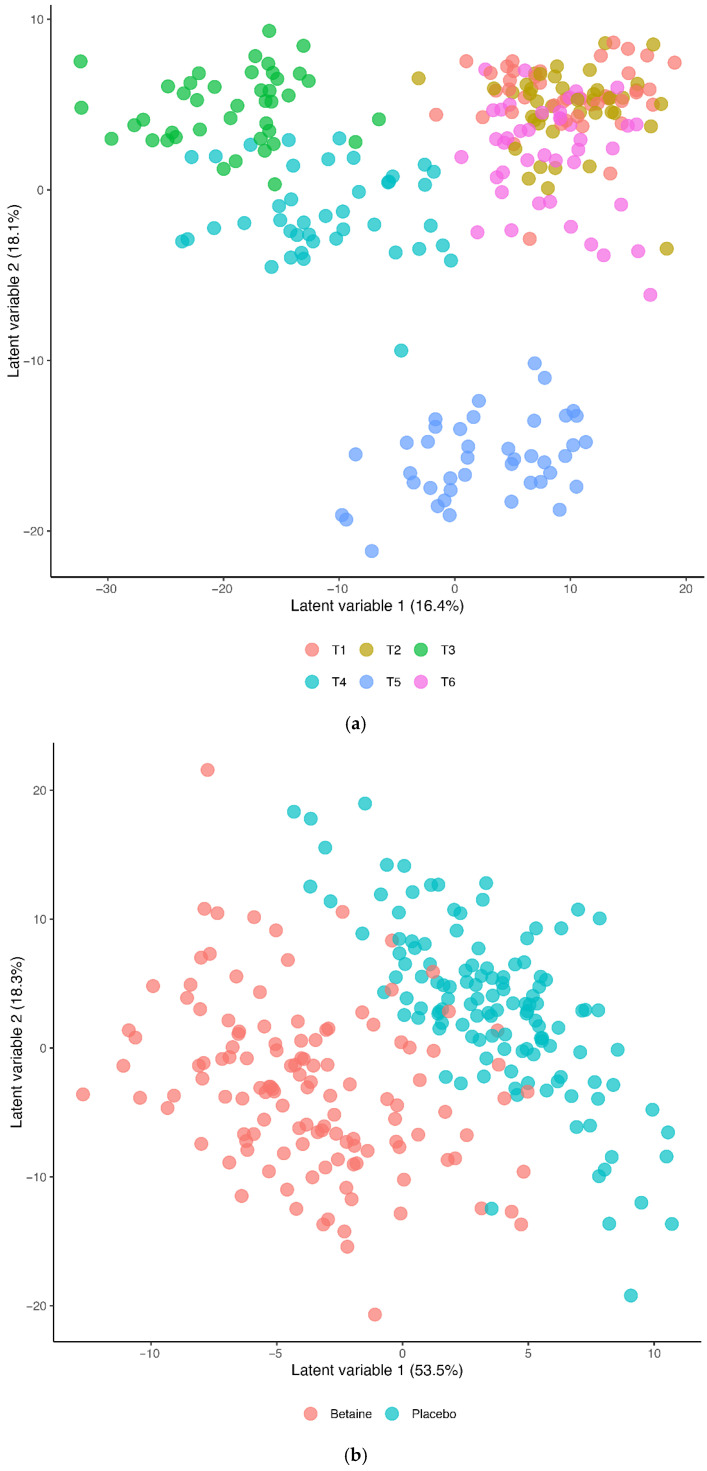
(**a**) Partial Least Squares Discriminant Analysis (OPLS-DA) for both trials across all six timepoints (T1 = presupplementation, T2 = post-2-weeks supplementation, T3 = immediately post-exercise, T4 = 1.5 h post-exercise, T5 = 3 h post-exercise, T6 = 24 h post-exercise). The 60 km cycling time trial had a significant effect on metabolite shifts (R2Y = 0.495, Q2 = 0.451). (**b**) OPLS-DA contrasting the betaine and placebo trials (all 6 timepoints). A distinct difference was shown between the betaine and placebo trials (R2Y = 0.769, Q2 = 0.574). R2Y values attributed to the contributory latent variables are listed for each axis in parentheses.

**Figure 5 nutrients-17-02765-f005:**
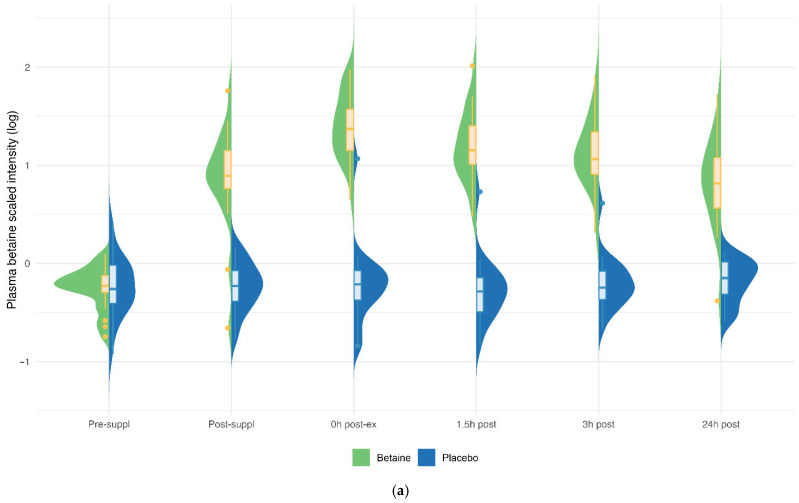
Betaine and placebo trial changes across six timepoints for plasma betaine (**a**), sarcosine (**b**), and dimethylglycine (DMG) (**c**). The treatment and timepoint interaction effects were each q < 0.001. The split violin graphs reveal the differences between plasma metabolite values for both trials across all six timepoints. Overlayed box-and-whisker plots further describe the minimum, maximum, and median values, the interquartile range, and detected outliers.

**Figure 6 nutrients-17-02765-f006:**
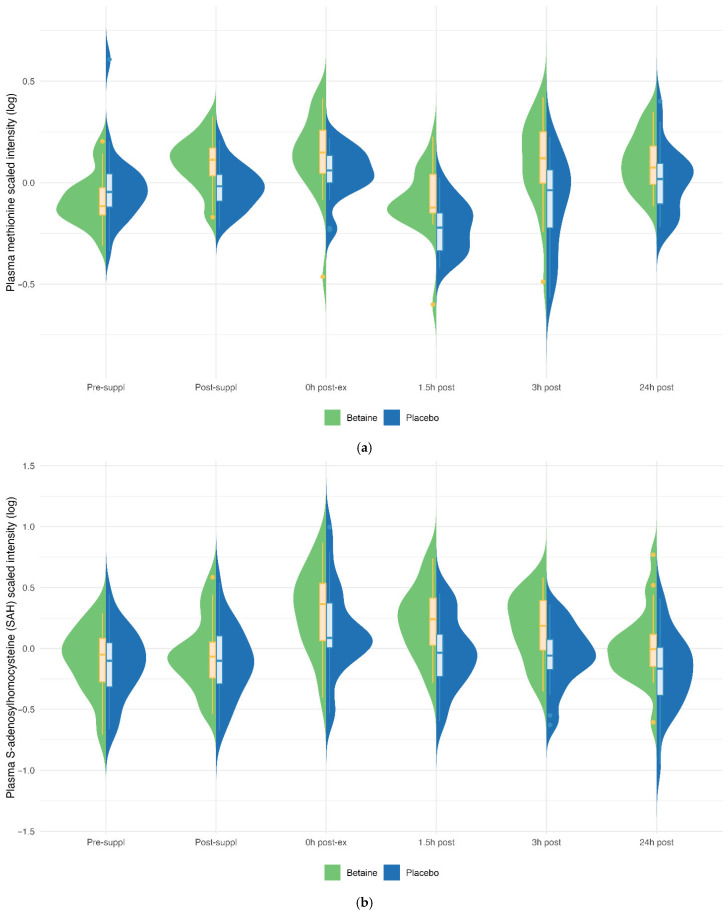
Betaine and placebo trial changes across six timepoints for plasma methionine (**a**), S-adenosylhomocysteine (SAH) (**b**), alpha-ketoglutaramate (**c**), and 5-methylthioadenosine (MTA) (**d**). The treatment and timepoint interaction effects were q = 0.036, 0.025, q < 0.001, and q < 0.001, respectively.

**Figure 7 nutrients-17-02765-f007:**
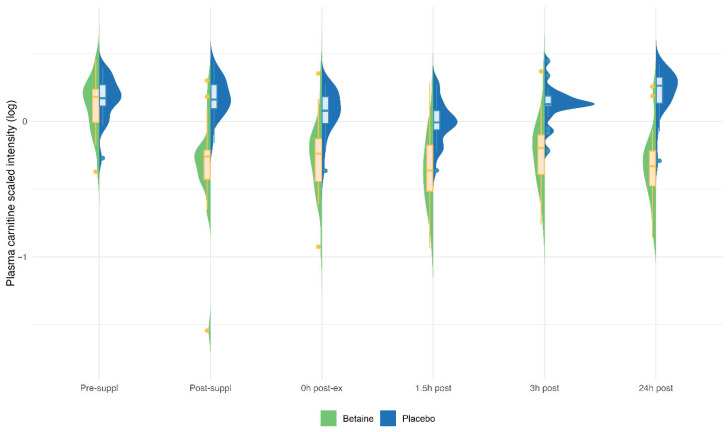
Betaine and placebo trial changes across six timepoints for plasma carnosine (q < 0.001).

**Table 1 nutrients-17-02765-t001:** Study participant characteristics for male and female cyclists (*n* = 21). * *p* < 0.05.

Variable	Males (*n* = 15)	Females (*n* = 6)
Age (years)	45.5 ± 2.5	44.2 ± 4.8
Weight (kilograms)	79.5 ± 2.4 *	62.5 ± 2.8
Height (centimeters)	180 ± 1.6 *	167 ± 3.6
Body mass index (BMI)	24.5 ± 0.5	22.6 ± 1.4
Body fat %	21.3 ± 1.7	23.5 ± 1.9
V0_2max_ (mL·kg^−1^min^−1^)	43.6 ± 1.6 *	36.7 ± 1.7
Maximal cycling power (watts)	268 ± 11.6 *	179 ± 15.0
Maximal heart rate (beats/min)	167 ± 3.0	162 ± 3.7
Maximal ventilation rate (liters/min)	126 ± 3.9 *	89.4 ± 9.9

**Table 2 nutrients-17-02765-t002:** The 60 km cycling performance data for *n* = 21 cyclists in the placebo and betaine trials (mean ± SE).

Performance Variable	Supplement	Mean ± SE	*t*-Test *p*-Value
Time to complete 60 km cycling trial (min)	Placebo Betaine	114.2 ± 2.6 112.8 ± 2.3	0.049
Average speed (km/h)	Placebo Betaine	31.8 ± 0.7 32.1 ± 0.6	0.132
Average watts and %maximal	Placebo Betaine	148 ± 9.1; 61.0 ± 1.9 147 ± 9.3; 60.5 ± 1.9	0.692; 0.674
Average VO_2_ (mL·kg^−1^·min^−1^) and %maximal	Placebo Betaine	30.6 ± 6.4; 74.6 ± 8.1 30.9 ± 6.4; 75.3 ± 8.5	0.327; 0.291
Average respiratory exchange ratio (RER) (VCO_2_/VO_2_)	Placebo Betaine	0.840 ± 0.007 0.838 ± 0.007	0.796
Average heart rate (bpm) and %maximal	Placebo Betaine	137 ± 3.9; 82.3 ± 1.7 138 ± 3.6; 82.9 ± 1.6	0.640; 0.608
Average rating of perceived exertion (RPE)	Placebo Betaine	13.7 ± 0.3 13.5 ± 0.3	0.412

**Table 3 nutrients-17-02765-t003:** Trial comparisons for *n* = 21 participants across all timepoints for intracellular water, total mood disturbance (TMD) from the Profile of Mood States (POMS) questionnaire, and physiological outcomes. *p*-values represent time (first value) and trial x time interaction effects.

Variable	Trial	Pre- Study	2-Wks Suppl.	0 h Post-Ex	1.5 h Post-Ex	3 h Post-Ex	24 h Post-Ex	*p*- Value
Intracellular water (liters)	Placebo	25.1 ± 1.0	25.2 ± 1.0	25.6 ± 1.0	25.0 ± 1.0	25.2 ± 1.0	25.1 ± 1.0	0.004; 0.010
	Betaine	25.6 ± 1.0	25.0 ± 1.0 *	25.2 ± 1.0 *	25.1 ± 1.0 *	25.0 ± 1.0 *	25.2 ± 1.0	
Total mood disturbance	Placebo	91.2 ± 2.2	94.3 ± 2.1	101 ± 2.2	98.5 ± 2.3	96.8 ± 2.0	91.8 ± 2.0	<0.001; 0.304
	Betaine	92.2 ± 1.7	90.4 ± 2.1	98.0 ± 2.3	96.0 ± 2.3	93.9 ± 1.9	90.7 ± 1.8	
DOMS (1–10 scale)	Placebo	1.9 ± 0.2	1.8 ± 0.2	4.5 ± 0.4	4.3 ± 0.5	3.9 ± 0.4	2.8 ± 0.4	<0.001; 0.465
	Betaine	1.7 ± 0.2	1.8 ± 0.2	4.1 ± 0.5	3.7 ± 0.4	3.0 ± 0.3	2.4 ± 0.4	
Creatine kinase (U/L)	Placebo	253 ± 61.7	192 ± 35.2	224 ± 37.9	215 ± 34.9	235 ± 38.2	254 ± 65.7	<0.001; 0.790
	Betaine	175 ± 22.3	152 ± 21.4	182 ± 24.2	178 ± 23.3	208 ± 27.1	233 ± 37.6	
Myoglobin (ng/mL)	Placebo	57.1 ± 17.3	43.8 ± 10.0	56.9 ± 5.2	107 ± 26.2	105 ± 39.3	36.9 ± 2.8	<0.001; 0.849
	Betaine	39.5 ± 2.8	33.9 ± 2.3	55.3 ± 5.1	96.7 ± 18.0	94.8 ± 22.2	38.2 ± 2.9	
Serum glucose (mg/dL)	Placebo	96.2 ± 2.1	95.0 ± 1.6	92.7 ± 3.3	84.4 ± 2.7	116 ± 4.8	93.0 ± 3.0	<0.001; 0.391
	Betaine	95.7 ± 2.0	92.6 ± 1.9	96.3 ± 3.0	85.6 ± 2.5	113 ± 5.0	94.4 ± 1.9	
Neutrophil/lymphocyte	Placebo	1.9 ± 0.1	1.8 ± 0.2	4.3 ± 0.3	6.7 ± 0.7	6.3 ± 0.5	2.0 ± 0.2	<0.001; 0.107
	Betaine	2.0 ± 0.2	1.7 ± 0.1	4.5 ± 0.4	7.2 ± 0.7	6.6 ± 0.8	2.2 ± 0.3	
Cortisol (µg/dL)	Placebo	17.3 ± 1.2	18.4 ± 1.2	21.5 ± 1.5	14.9 ± 1.3	11.4 ± 1.3	16.2 ± 1.5	<0.001; 0.955
	Betaine	17.3 ± 1.1	18.0 ± 1.1	21.8 ± 1.7	15.1 ± 1.4	11.0 ± 0.9	15.8 ± 1.1	

* *p* < 0.0125 test for change between trials; Ex = exercise; Suppl = supplement; DOMS = delayed onset of muscle soreness.

## Data Availability

The raw data supporting the conclusions of this article will be made available by the authors, without undue reservation. The metabolomics data has been deposited to the MetaboLights repository (https://www.ebi.ac.uk/metabolights/MTBLS976) (accessed 15 March 2025) with the dataset identifier MTBLS976.

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
