# Peer review of "Betaine Supplementation Improves 60 km Cycling Time Trial Performance and One-Carbon Metabolism in Cyclists During Recovery"

_nutrients, 2025, doi:10.3390/nu17172765_

Round 1

Reviewer 1 Report

Comments and Suggestions for Authors

The manuscript reports a crossover trial of betaine supplementation (3 g/day for 2 weeks) in cyclists. Based on their findings, the authors report that betaine improved 60 km time trial performance and altered one-carbon metabolism. The performance effect is marginal (effect size=0.457, p = 0.049) and perhaps of questionable practical relevance. Most of the secondary endpoints are null.

  • The paper addresses an underexplored topic and is of relevance to the scope of the journal and the special issue
  • The authors integrate performance metrics with biochemical data and metabolomics, which is an innovative and data-rich approach for sports science
  • Extensive background on betaine’s biochemical roles and prior research in both human and animal modelsis presented
  • The crossover study design is the optimal choice in such cases
  • The dose and the duration of the supplementation period are well justified
  • The methodology and the metabolomics data are comprehensively described
  • Trial registration and ethical approval are both documented
  • The paper is well structured and written

Please find below some comments on the paper, that may be of help:

  1. Given, the borderline primary significant finding, and the small absolute gain in performance, it would be useful if authors could justify its competitive or physiological importance.
  2. Generally, I feel that the authors could partially temper the performance-enhancing claims to reflect borderline significance, given that all other performance indices were not significant (Table 2).
  3. Please provide the confidence intervals for the effect size (0.457).
  4. The last dose of betaine was ~20min before the 60-km trial (7:10 and 7:30, respectively). How can one be sure that the performance effects, and especially, the metabolomics data are due to the chronic supplementation protocol and not due to the last dose few hours before?
  5. Since no direct epigenetic, transcriptional, or methylation data were collected, the largest part of the discussion remains speculative (i.e., mechanistic causality via one-carbon metabolism).
  6. Most metabolomics “findings” actually reflect ingestion/metabolism of betaine itself. Are there any other molecules that could be related to performance-associated pathways?
  7. My apologies if I lost something in the interpretation, but was the gut permeability challenge indeed stressful enough to observe any betaine-induced effect compared to placebo (what was the effect of the challenge in the placebo condition)?
  8. Did the authors apply a multiplicity control strategy for the large number of analyses?
  9. Diet control relied solely on self-report; biomarker validation is absent – potential study limitation?

Author Response

Reviewer #1

The manuscript reports a crossover trial of betaine supplementation (3 g/day for 2 weeks) in cyclists. Based on their findings, the authors report that betaine improved 60 km time trial performance and altered one-carbon metabolism. The performance effect is marginal (effect size=0.457, p = 0.049) and perhaps of questionable practical relevance. Most of the secondary endpoints are null.

  • The paper addresses an underexplored topic and is of relevance to the scope of the journal and the special issue
  • The authors integrate performance metrics with biochemical data and metabolomics, which is an innovative and data-rich approach for sports science
  • Extensive background on betaine’s biochemical roles and prior research in both human and animal models is presented
  • The crossover study design is the optimal choice in such cases
  • The dose and the duration of the supplementation period are well justified
  • The methodology and the metabolomics data are comprehensively described
  • Trial registration and ethical approval are both documented
  • The paper is well structured and written

RESPONSE:  Thank you for taking the time and effort to review our paper.

Please find below some comments on the paper, that may be of help:

  1. Given, the borderline primary significant finding, and the small absolute gain in performance, it would be useful if authors could justify its competitive or physiological importance… Generally, I feel that the authors could partially temper the performance-enhancing claims to reflect borderline significance, given that all other performance indices were not significant (Table 2).

RESPONSE: In the abstract and discussion, we characterize the 60-km cycling performance improvement of 1.41 minutes as modest. The effect size of 0.457 falls into the medium effect size category. Cohen's d thresholds are 0.2 (small), 0.5 (medium), and 0.8 (large). Thus, the observed trial difference is considered moderate, but not necessarily very strong. We added a statement in the second paragraph summarizing this information (in response to your comment). We also tempered the performance claim in the first sentence of the conclusion section.

  1. Please provide the confidence intervals for the effect size (0.457).

RESPONSE: The confidence intervals for the effect size have been added in the results section (in response to your request).

  1. The last dose of betaine was ~20min before the 60-km trial (7:10 and 7:30, respectively). How can one be sure that the performance effects, and especially, the metabolomics data are due to the chronic supplementation protocol and not due to the last dose few hours before?

RESPONSE: As explained in the last paragraph of the introduction, our 3 g/d, 2-week dosing protocol was based on published data regarding plasma betaine and homocysteine (reference #6). We agree that we cannot rule out an acute effect on outcome measurements due to the pre-exercise dose of betaine. We added a statement in the limitations paragraph (final paragraph of the discussion): “We cannot rule out that the performance effect and metabolite shifts reported in this study were due in part to the acute intake of betaine just prior to the cycling time trial.”

  1. Since no direct epigenetic, transcriptional, or methylation data were collected, the largest part of the discussion remains speculative (i.e., mechanistic causality via one-carbon metabolism).

RESPONSE: We agree and listed this as a limitation (last paragraph of the discussion). We do feel that the metabolomics data showing an increase in the DNA methyl donor pool justifies discussing the potential for epigenetic DNA methylation and that this is an area for future research.

  1. Most metabolomics “findings” actually reflect ingestion/metabolism of betaine itself. Are there any other molecules that could be related to performance-associated pathways?

RESPONSE: As emphasized in the results section, the PLS-DA and Lasso regression analyses identified just 9 metabolites that were most influential in distinguishing differences between the betaine and placebo trials. As you emphasized, these were related to betaine ingestion. We did not find any metabolites that differed between the trials that could be regarded as performance related. As mentioned in the discussion (paragraph 5), our RER data did not support a glycogen-sparing effect due to betaine-related effects on carnitine.

  1. My apologies if I lost something in the interpretation, but was the gut permeability challenge indeed stressful enough to observe any betaine-induced effect compared to placebo (what was the effect of the challenge in the placebo condition)?

RESPONSE: As emphasized in paragraph 7, “A limitation in this study was that changes in gut permeability were modest following the 60-km cycling bout.” One review concluded that vigorous endurance exercise lasting at least 60 min at 70% VO2max was a sufficient stimulus to induce increased intestinal permeability, but that this varies widely between participants (Ribeiro, F.M.; Petriz, B.; Marques, G.; Kamilla, L.H.; Franco, O.L. Is there an exercise-intensity threshold capable of avoiding the leaky gut? Front. Nutr. 2021, 8, 627289). We added this statement and reference to the paper in paragraph 7 (in response to your comment).

  1. Did the authors apply a multiplicity control strategy for the large number of analyses?

RESPONSE: Yes, a p-value adjustment was applied using Benjamini-Hochberg (see line 314).

  1. Diet control relied solely on self-report; biomarker validation is absent – potential study limitation?

RESPONSE: We agree that diet intake data was based on self-reported records. However, this was a randomized controlled trial using a crossover design. Thus, our statement in the results section that “no differences in macro- and micro-nutrient intake were found between trials” is acceptable, we believe.

Reviewer 2 Report

Comments and Suggestions for Authors

The study is well designed, with a clear randomized, double-blind, crossover protocol, and the methodology is described in detail. The introduction provides a comprehensive background with relevant and recent references. The metabolomics approach adds novelty, and the dietary control and washout period strengthen the reliability of the findings. 

However, the performance improvement, although statistically significant, is modest and may have limited practical relevance for trained cyclists. The absence of effects on most secondary outcomes (gut permeability, muscle damage, inflammation, cortisol, mood) should be emphasized more clearly in the discussion, as well as the unexpected decrease in intracellular water. The mechanistic link with one-carbon metabolism is plausible but remains speculative without direct measures of methylation or gene expression. 

Beware that there are several parts with plagiarism (check the mdpi report).

The discussion would benefit from:

– Placing the magnitude of performance change in the context of competitive relevance.

– Addressing why the hypothesized gut permeability effects were not observed, and considering environmental or exercise intensity factors.

– Further exploring potential explanations for the reduction in intracellular water.

Figures and tables are clear and support the text. Overall, this is a solid study with interesting findings that merit publication, but some claims in the discussion should be more cautious given the modest effect size and lack of changes in key physiological markers.

Author Response

Reviewer #2

The study is well designed, with a clear randomized, double-blind, crossover protocol, and the methodology is described in detail. The introduction provides a comprehensive background with relevant and recent references. The metabolomics approach adds novelty, and the dietary control and washout period strengthen the reliability of the findings. 

RESPONSE:  Thank you for taking the time and effort to review our paper.

However, the performance improvement, although statistically significant, is modest and may have limited practical relevance for trained cyclists. The absence of effects on most secondary outcomes (gut permeability, muscle damage, inflammation, cortisol, mood) should be emphasized more clearly in the discussion, as well as the unexpected decrease in intracellular water. The mechanistic link with one-carbon metabolism is plausible but remains speculative without direct measures of methylation or gene expression. 

RESPONSE: See responses below. Also, we added this statement to the conclusion (in response to your comment regarding secondary outcomes):

“Betaine supplementation did not influence several post-exercise secondary outcomes in these trained cyclists including gut permeability, muscle damage, inflammation, serum cortisol levels, or mood state.”

Beware that there are several parts with plagiarism (check the mdpi report).

RESPONSE: We used methods that were previously described in similar prior studies. This is not plagiarism, we believe.

The discussion would benefit from:

– Placing the magnitude of performance change in the context of competitive relevance.

RESPONSE: We added this statement to the second paragraph of the discussion:

“The effect size of 0.457 falls into the medium category meaning that the observed cycling trial performance difference is evident, but not necessarily very strong.”

– Addressing why the hypothesized gut permeability effects were not observed, and considering environmental or exercise intensity factors.

RESPONSE: As emphasized in paragraph 7, “A limitation in this study was that changes in gut permeability were modest following the 60-km cycling bout.” One review concluded that vigorous endurance exercise lasting at least 60 min at 70% VO2max was a sufficient stimulus to induce increased intestinal permeability, but that this varies widely between participants (Ribeiro, F.M.; Petriz, B.; Marques, G.; Kamilla, L.H.; Franco, O.L. Is there an exercise-intensity threshold capable of avoiding the leaky gut? Front. Nutr. 2021, 8, 627289). We added this statement and reference to the paper in paragraph 7 in response to your comment.

– Further exploring potential explanations for the reduction in intracellular water.

RESPONSE: We have a full paragraph in the discussion on this issue. We have explored the literature carefully and added this statement to the bottom of paragraph 6 in the discussion (in response to your comment): “Further research is needed using sophisticated fluid balance and diet control methods to determine the potential effects of betaine supplementation on intracellular water and performance in human athletes.”

Figures and tables are clear and support the text. Overall, this is a solid study with interesting findings that merit publication, but some claims in the discussion should be more cautious given the modest effect size and lack of changes in key physiological markers.

RESPONSE: Thank you and we have attempted to make the changes you recommended.

Round 2

Reviewer 2 Report

Comments and Suggestions for Authors

The authors have made some improvements to the paper.

A plagiarism check, see attached file, shows that many parts are palgiated.

the paper cannot be accepted in this form.

Author Response

We added the reference listed below and stated that methods related to the study design, exercise protocol, and gut permeability were similar or adapted for this study.  (See methods section).

24. Nieman DC, Sakaguchi CA, Williams JC, Pathmasiri W, Rushing BR, McRitchie S, Sumner SJ. Selective Influence of Hemp Fiber Ingestion on Post-Exercise Gut Permeability: A Metabolomics-Based Analysis. Nutrients. 2025 Apr 19;17(8):1384. doi: 10.3390/nu17081384. PMID: 40284247; PMCID: PMC12030204.

References following #24 have been renumbered.

We made a minor correction to the effect size statistics in the results section (brought to my attention by the statistician). This did not change the meaning of the results.